# Cellulose Isolation from Tomato Pomace Pretreated by High-Pressure Homogenization

**DOI:** 10.3390/foods11030266

**Published:** 2022-01-19

**Authors:** Annachiara Pirozzi, Giovanna Ferrari, Francesco Donsì

**Affiliations:** 1Department of Industrial Engineering, University of Salerno, Via Giovanni Paolo II, 132, 84084 Fisciano, SA, Italy; apirozzi@unisa.it (A.P.); gferrari@unisa.it (G.F.); 2ProdAl Scarl, University of Salerno, Via Giovanni Paolo II, 132, 84084 Fisciano, SA, Italy

**Keywords:** tomato pomace, cellulose, acid hydrolysis, alkaline hydrolysis, bleaching, high-pressure homogenization, defibrillation, bioactive compounds

## Abstract

This work proposes a biorefinery approach for the utilization of agri-food residues, such as tomato pomace (TP), through combining chemical hydrolysis with high-pressure homogenization (HPH), aiming to achieve the isolation of cellulose with tailored morphological properties from underused lignocellulose feedstocks, along with the valorization of the value-added compounds contained in the biomass. Cellulose was isolated from TP using sequential chemical hydrolysis in combination with mechanical pretreatment through HPH. The chemical and structural features of cellulose isolated from TP pretreated by HPH were compared with cellulose isolated from untreated TP through light scattering for particle size distribution, optical and scanning electron microscopy, and Fourier-transform infrared spectroscopy (FT-IR) analysis. HPH pretreatment (80 MPa, 10 passes) not only promoted a slight increase in the yield of cellulose extraction (+9%) but contributed to directly obtaining defibrillated cellulose particles, characterized by smaller irregular domains containing elongated needle-like fibers. Moreover, the selected mild chemical process produced side streams rich in bioactive molecules, evaluated in terms of total phenols and reducing activity. The liquors recovered from acid hydrolysis of TP exhibited a higher biological activity than those obtained through a conventional extraction (80% *v*/*v* acetone, 25 °C, 24 h at 180 rpm).

## 1. Introduction

In recent years, cellulose has received increasing interest as a renewable raw material for producing biodegradable polymeric products and contributing to replacing fossil resources, considering their depletion, fluctuation in oil prices, and the negative environmental impacts [1]. Cellulose is a linear homopolymer of glucose (C_6_H_12_O_6_)_n_, consisting of repeated units of D-glucose in a ^4^C_1_ conformation. In the global scenario, among the available sources of renewable feedstock for cellulose isolation, lignocellulosic biomass, including agricultural residues, stands out. In contrast, the use of the vastly available by-products and wastes, also known as agri-food residues (AFR), is currently limited to livestock feed, providing a limited added value, or landfill or energy production by combustion, causing potential environmental issues [2]. Therefore, new strategies for the exploitation of AFR represent a great opportunity for more sustainable routes for the recovery of high value-added compounds, as well as for mitigating their environmental burden.

To date, research has primarily focused on the production of cellulose from lignocellulosic waste, such as coconut husk fibers [3], cassava bagasse [4], hazelnut shells [5], rice husk [6,7,8], wheat straw [9,10,11], oat hull [12], and okara [13,14]. All these biomasses mainly consist of three natural organic polymers, cellulose, hemicellulose, and lignin, and also contain small amounts of proteins, pectin, and other extractives (e.g., bioactive molecules). The cellulose isolation process is typically based on strong acid hydrolysis (such as 63–72% H_2_SO_4_ [15,16], 42% HCl [17,18] or 77–83% H_3_PO_4_ [19]), followed by alkaline hydrolysis (1–10% NaOH [20,21,22]) and a bleaching process to remove residual pigments (1–2% NaClO [23,24,25] or 5–15% H_2_O_2_ [26,27]). Milder processing (e.g., reducing acid and alkali concentrations) conditions are desirable for more sustainable cellulose production processes. However, available data show intrinsic limitations in the cellulose extraction that can be obtained: lower acid concentrations are reported to produce particles characterized by highly amorphous regions with low crystallinity, therefore, reducing the yield and purity of cellulose [28].

The tomato processing industry represents an interesting case study for enhancing and integrating synergic solutions for waste management, considering that tomato is one of the most widely cultivated vegetable crops in Mediterranean countries. It is one of the largest primary vegetable crops. As such, the tomato processing industry is responsible for the production of large amounts of solid waste (~3–4% of the fresh processed tomatoes weight, also known as tomato pomace, TP), which creates a major disposal problem in terms of costs and environmental impacts [29]. Nowadays, TP is partially exploited for the organic extraction of lycopene [30,31], a powerful natural antioxidant carotenoid widely used in food, pharmaceutical, and cosmetic products [32]. More recently, TP has been investigated as feedstock for anaerobic digestion to produce biogas [30,33,34]. However, TP, consisting of peels, seeds, and fibrous residues, can also be exploited to recover cellulose because it is a good source of the complex carbohydrates composing the lignocellulosic plant cell wall (approximately 65% on a dry mass basis of TP) [35].

In general, cellulose isolation from AFR requires a top-down chemical deconstructing strategy, conventionally based on acid or enzymatic hydrolysis. Acid hydrolysis is the most widely used strategy because it is inexpensive and can induce the breakdown of the fibrous cell walls, remove the lignin and hemicellulose fraction, and, thus, expose the cellulose for extraction [36]. However, there are several disadvantages in using this conventional approach, such as corrosion of equipment, environmental damage, as well as safety issues [13]. For these reasons, the increase in efficiency of these isolation methods, while decreasing the severity of the operating conditions, is highly desirable.

In this scenario, the aim of this study is to develop an efficient process to isolate cellulose from TP as a representative AFR via the combination of mechanical treatments by high-pressure homogenization technology (HPH) and mild chemical hydrolysis processes. The fluid-mechanical stresses on the biomass occurring within the interaction chambers during HPH treatment were previously reported to induce (i) a significant reduction in particle size, (ii) an increase in surface area [37], as well as (iii) the disruption of the well-arranged cellulose-hemicellulose-lignin complexes, causing what is referred to as fiber activation or cellulose defibrillation [38]. The HPH fluid-mechanical treatment is here exploited for the first time to increase the efficiency of mild chemical hydrolysis processing for isolation of cellulose with a defibrillated structure from tomato pomace. Moreover, it is also aimed to demonstrate that the selected mild chemical hydrolysis conditions are able to produce side streams very rich in bioactive compounds, further contributing to the valorization of the agri-food residue under study.

## 2. Materials and Methods

### 2.1. Tomato Pomace

Tomato pomace (TP) from industrial processing, mainly composed of skins and seeds, was kindly provided by Salvati Mario & C. spa (Mercato San Severino, Italy). The tomato pomace was dried in an oven at 50 °C for 48 h. The dried material was milled in a lab knife grinder and sieved to a final particle size ≤ 2 mm. The milled pomace (sample TP) was packed in vacuum-sealed flexible pouches (multilayer packaging OPP30-A19-LDPE70, Di Mauro Officine Grafiche S.p.A., Cava de’ Tirreni, Italy) and kept at 5 °C until used.

### 2.2. Chemicals

Sulfuric acid (H_2_SO_4_, 95.0–98.0%, ACS GR, Fluka, Charlotte, NC, USA), sodium hydroxide (NaOH, beads, PanReac, Barcelona, Spain), hydrogen peroxide (H_2_O_2_, 6% solution, ACS GR, VWR Chemicals, Radnor, PA, USA), Folin-Ciocaleau’s Reagent (1.8–2.2 mol/L, PanReac, Barcelona, Spain), sodium carbonate (NaCO_3_, ACS GR, PanReac, Barcelona, Spain), TPTZ (2,4,6-Tris (2-pyridil)-s-triazine, ≥99.0%, Sigma-Aldrich, St. Louis, MO, USA), chloridric acid (HCl, 36.5–38.0%, ACS GR, PanReac, Barcelona, Spain), iron (III) chloride hexahydrate (FeCl_3_·6H_2_O, ≥98%, Sigma-Aldrich, St. Louis, MO, USA), acetone ((CH_3_)_2_CO, ≥99.0%, VWR Chemicals, Radnor, PA, USA), ethanol (C_2_H_5_OH, 99.9%, VWR Chemicals, Radnor, PA, USA), methanol (CH_3_OH, ≥95%, Thermo Scientific, Waltham, MA, USA) were used as received without further purification. All water used throughout this work was purified by a Milli-Q water purification system (Barnstead™ Pacific TII Water, Thermo Scientific, Waltham, MA, USA).

### 2.3. High-Pressure Homogenization Treatment of Tomato Pomace

The fresh tomato pomace, milled in a lab knife grinder and suspended in bidistilled water (5 g/L of dry weight), was subjected to high-shear mixing (HSM) at 20,000 rpm for 10 min with a T-25 Ultra Turrax device (IKA^®^-Werke GmbH & Co. KG, Staufen, Germany) equipped with an S25-N18 G rotor in an ice bath, to avoid any temperature rises. Subsequently, the obtained suspension was treated by HPH using an orifice valve assembly (orifice diameter of 200 μm) at 80 MPa and 25 °C for up to 10 passes, using an in-house developed system described in detail elsewhere [37]. At the end of the HPH treatment, the suspension was concentrated by using an R-200/205 Rotavapor (BÜCHI Labortechnik AG, Flawil, Switzerland) until achieving a volume reduction of up to 70%, prior to being freeze-dried in a 25 L VirTis Genesis freeze-drier (SP Scientific Products, Stone Ridge, NY, USA) at 7 Pa for 72 h. The HPH freeze-dried extract (sample HPH-TP) was then packed in vacuum-sealed flexible pouches and stored under refrigerated conditions before proceeding to the isolation of cellulose through the chemical hydrolysis steps (Section 2.6).

### 2.4. Chemical Characterization of Raw Materials

The composition of TP samples and high-pressure homogenized TP (HPH-TP) was determined after applying the following official analytical methods. Moisture and ash content were evaluated by drying at 105 °C in an air-force oven (950.46 AOAC) [39] and at 525 °C in a muffle (923.02 AOAC) [39], respectively. Total protein content was determined by the Kjeldahl method (954.01 AOAC) [39]. Total fat was quantified by gravimetry after extraction with a mixture of petroleum ether and diethyl ether (1:1) (920.39 AOAC) [39].

The extractives, cellulose, hemicelluloses, and lignin content were determined using the gravimetric method according to the methods reported by the Technical Association of Pulp and Paper Industry (TAPPI). Briefly, during the first isolation step, the extractive content was determined through the TAPPI method (T-264 om-82) [40] and ASTM E1721-01 using a Soxhlet apparatus to remove sugars, phenolic compounds, and part of water-soluble polysaccharides. Lignin extraction was based on the TAPPI T-222 om-22 methodology [41], according to which the obtained extractive-free sample was subjected to acid hydrolysis. Cellulose extraction, based on the TAPPI T-203 method, was obtained by alkaline treatment with soda and acetic acid applied on extractive-free samples from the first isolation step.

### 2.5. Optimization of Bioactives Extraction

A traditional extraction process with organic solvents was performed to recover the bioactive compounds from TP and HPH-TP at a fixed liquid-to-solid ratio equal to 10 under continuous agitation in a thermostated orbital shaker at 180 rpm for 24 h at 25 ± 1 °C. Three green solvents (acetone, ethanol, and methanol) at different dilutions with distilled water were tested to identify the optimal conditions for the total phenols extraction. Response Surface Methodology (RSM) for the design of experiments (DOE) was used to optimize the operating parameters. The type of solvent (discrete numeric factor) and solvent concentration (from 20 to 100 % *v*/*v*) were selected as the independent variables to be optimized. The response variable was the concentration of total phenols extracted, evaluated through the Folin-Ciocalteau method following the methodology described in Section 2.7.1.

I-Optimal Design was applied in this study, using Design-Expert 11.1.2.0 software (Statease Inc., Minneapolis, MN, USA) to determine the number of experiments to be evaluated for the optimization of the two independent and the response variables (Table 1).

### 2.6. Isolation of Cellulose

Cellulose was extracted from TP and HPH-TP through a chemical route, consisting of three consecutive steps (Figure 1). Initially (first step), the samples were treated with a 4.7% *v*/*v* H_2_SO_4_ solution (1:10 m_Sample_:vs_olution_) in a static autoclave at 121 °C for 45 min to hydrolyze polysaccharides and acid-soluble polyphenols. After acid hydrolysis, the samples were removed from the autoclave and rapidly cooled under running water. The second step consisted of the alkaline hydrolysis carried out by using 4 N NaOH solution (1:10 m_Sample_:vs_olution_) at 25 °C for 24 h under continuous stirring to dissolve the remaining hemicellulose, lignin, and other polysaccharides. Finally, in the third step, the samples were bleached with 4% H_2_O_2_ solution, with pH adjusted to 11.5 with NaOH (1:10 m_Sample_:vs_olution_), at 45 °C for 8 h under continuous stirring. At the end of each step, the solid residue was collected by vacuum filtration, washed by flushing distilled water until the pH of the eluted water reached neutral values, and dried in an oven at 50 °C for 24 h. The resulting cellulose pulp was dried to be used for further characterizations.

### 2.7. Liquor Analysis

#### 2.7.1. Total Phenolic Content

Total phenols were quantified through the Folin-Ciocalteau assay [42] adopted to study natural antioxidant compounds; it measures the capacity of a compound to reduce the Folin’s reagent, giving an estimation of its antioxidant capacity. Briefly, the liquor sample (1 mL opportunely diluted with distilled water) was added to Folin-Ciocalteau’s reagent (5 mL at 10% *v*/*v*) and allowed to stand for 5 min at room temperature. Then, 7.5% *w*/*v* sodium carbonate solution (4 mL) was added to the mixture. After vortexing for 1 min, the mixture was incubated at room temperature for 60 min in darkness. The absorbance of the reacting mixture was then read at 765 nm using a V-650 Ultraviolet–visible (UV–Vis) spectrophotometer (Jasco Inc., Easton, MD, USA). Total phenols were expressed as milligrams of gallic acid equivalents per gram of dry samples (mg_GAE_/g_DM_), by means of a calibration curve obtained with a gallic acid standard at different concentrations (10–100 mg/L).

For the detection of the total content of phenolic compounds in liquors, the 280-index method was also used: the sample was opportunely diluted with distilled water, and absorbance was directly read at 280 nm, which is the characteristic absorption wavelength of a benzene ring. The calibration curve was obtained with gallic acid standard solutions (5–20 mg/L).

#### 2.7.2. Reducing Activity

The reducing capacity of liquors was evaluated with the FRAP (ferric reducing antioxidant power) assay [43]. The FRAP working solution was prepared by freshly mixing 0.3 M sodium acetate buffer, 10 mM TPTZ (2,4,6-Tris (2-pyridyl)-s-triazine) solution in 40 mM HCl, and 20 mM ferric solution at a ratio of 10:1:1 (*v*/*v*/*v*). Freshly prepared FRAP reagent solution (2.5 mL) was mixed with 0.5 mL of an opportunely diluted sample and incubated for 10 min at ambient temperature. The absorbance was measured at 593 nm using a V-650 UV–Vis spectrophotometer (Jasco Europe Srl, Cremella, Italy) against blank (applying the same analysis conditions). Ascorbic acid was used as the standard for calibration curve, and the FRAP values were expressed as ascorbic acid equivalents (mg_AA_) present in the dry analyzed sample (g_DW_).

### 2.8. Cellulose Pulp Characterization

The yield of obtained cellulose (%) was calculated as the ratio between the weight of dried cellulose obtained from the third step of the cellulose isolation and the dried weight of pomace, as shown in Equation (1):(1)$$Yield\;\left(\%\right)=\frac{W_{Dried\;cellulose}}{W_{Dried\;pomace}}\cdot \;100$$

#### 2.8.1. Structural Carbohydrate and Lignin Analysis

The cellulose residues obtained from TP and HPH-TP after chemical treatments were subjected to strong acid hydrolysis followed by dilute acid hydrolysis for the quantification of cellulose, hemicellulose, and lignin contents, according to Sluiter et al. 2011 [44]. Briefly, the residues were treated with 72% H_2_SO_4_ solution (0.2:1 m_Sample_:v_Solution_) at 30 °C in a water bath for 60 min. Then, distilled water was added to obtain a final concentration in H_2_SO_4_ equal to 3% *v*/*v,* and the mixture was autoclaved at 121 °C for 60 min. After cooling down at room temperature, the solid fraction was separated from the liquid one by using Robu™ borosilicate glass filter crucible (Thermo Fisher Scientific, Waltham, MA, USA) with a porosity of 2 (40–100 µm pore size). The solid residue was analyzed for moisture (according to AOAC 950.46 [39]) and ash content (according to AOAC 923.02 [39]) to determine the acid-insoluble lignin fraction; meanwhile, the liquid fraction was analyzed for acid-soluble lignin, cellulose, hemicellulose, and acetic acid. More in detail: acid-soluble lignin was determined by reading the absorbance at 320 nm in a V-650 UV-Vis spectrophotometer (Jasco Europe Srl, Cremella, Italy); cellulose content (expressed as glucan) was evaluated from the D-glucose concentration by using the GOPOD-format kit assay (K-Gluc assay, Megazyme, Lansing, MI, USA) multiplied by the correction factor of 0.88; hemicellulose concentration was quantified from the xylose concentration (K-Xylose assay, Megazyme, Lansing, MI, USA) multiplied by the correction factor of 0.90; acetic acid concentration was determined by using the acetate kinase/phosphotransacetylase (AK/PTA) format kit assay (K-Acetrm assay, Megazyme, Lansing, MI, USA).

#### 2.8.2. Fourier Transform Infrared Spectroscopy (FT-IR)

Fourier-transform infrared (FT-IR) spectroscopy was used to obtain the spectra of the cellulose samples, an FT-IR-4100 series spectrophotometer (Jasco Europe Srl, Cremella, Italy) at ambient conditions using a single-reflectance horizontal ATR cell (ATR-PRO 470-H with a diamond prism, Jasco Europe Srl, Cremella, Italy). The infrared spectra were collected in absorbance mode from an accumulation of 128 scans over the wavenumber regions of 4000–400 cm^−1^ at a resolution of 4 cm^−1^. Three repetitions from each sample were used for each spectra measurement. The resulting averaged spectra were smoothed with five-point under adaptive-smoothing function to remove the eventual noises, and then baseline modification was applied.

#### 2.8.3. Morphological Analysis

The morphology of celluloses was observed with an optical microscope (Nikon Eclipse TE 2000S, Nikon instruments Europe B.V., Amsterdam, The Netherlands) with a 10× objective, coupled to a DS Camera Control Unit (DS-5M-L1, Nikon Instruments Europe B.V, Amsterdam, The Netherlands) for image acquisition and analysis. For scanning electron microscope (SEM) observations, the samples were mounted on an aluminum stub and coated by a 10 nm thick gold-palladium alloy sputter coater before observation in a high-resolution ZEISS HD15 Scanning Electron Microscope (Zeiss, Oberkochen, Germany) at 20,000× magnification.

#### 2.8.4. Particle Size Distribution

The size distributions of TP and HPH-TP isolated cellulose particles were measured by laser diffraction using a Mastersizer 2000 instrument (Malvern instrument Ltd., Malvern, UK), as reported by Pirozzi et al. 2021 [45]. Characteristic diameters d(0.1), d(0.5), and d(0.9) were evaluated, corresponding to the 10th, 50th (median value), and 90th percentile of the cumulative size distribution of the suspensions.

### 2.9. Statistical Analysis

All experiments and analysis were performed in triplicate, otherwise specified, and the mean and standard deviation (SD) of the experimental values were calculated. Statistically significant differences among the averages were assessed by one-way ANOVA and the Tukey’s test (*p* < 0.05), using statistical software SPSS (version 20, SPSS Inc., Chicago, IL, USA).

## 3. Results and Discussion

### 3.1. Chemical Composition of TP and HPH-TP

The chemical composition of fresh tomato pomace and HPH-treated tomato pomace after freeze-drying is reported in Table 2.

The fluid-mechanical stresses involved in HPH may affect the content of food macromolecules (fat, proteins, and polysaccharides). The comparison of the ash content of the two samples provides evidence of mass conservation for the different treatments and about their comparability: the ash content of the fresh TP was not significantly different from the sample treated by HPH and freeze-dried. However, after HPH treatment, a decrease of 16.1% in protein content occurred. The HPH process is reported to contribute to increasing water solubility of proteins [46,47]; therefore, the higher release of water-soluble proteins due to HPH might have promoted their denaturation during the applied concentration and freeze-drying treatments. While the fat content did not significantly change during HPH treatment, an increase of 77.9% in extraction yield of phenolic acids and carotenoids (classified as extractives in Table 2) was observed for HPH-TP in comparison with TP. This suggests that HPH treatment is able to cause structural changes to the cell compartments by disruption of cell walls and membranes and promoting the release of bioactive intracellular compounds, which were measured in higher concentration, in very good agreement with previous results [37,48]. The applied HPH treatment also influenced the composition of the fiber by reducing the resistance between the cellulose chain and disrupting the intermolecular and intramolecular hydrogen bonds [49]. HPH caused a significant (*p* < 0.05) increase in cellulose content, which correlates well with the observed decrease in lignin content. The lignin-containing fibers are subjected to intense collision, shearing, and cavitation, and the fibers are more likely to break up into uniform defibrillated cellulose of micrometric size during the HPH treatment [50].

### 3.2. Optimization of Conventional Extraction Conditions

The effect of solvent type and solvent concentration during solid-liquid extraction with organic solvents were investigated to identify the highest yields of total phenols, using conventional methods under optimized conditions. RSM was used to identify the optimum values of independent variables (operating conditions). The interaction graphs in Figure 2 provide a method to visualize the relationship between the response variable (total phenols content) and solvent concentration for each solvent type. More details about the optimization of conventional extraction, RSM, and 3D contour plots are reported in the Appendix A.

Under the same extraction conditions, e.g., time, temperature, agitation speed, sample particle size, and solid-liquid ratio, the extraction efficiency is affected by the chemical nature of the phytochemicals, as well as the solvent used, for its polarity and concentration. However, in the case of vegetable biomass, the composition of the extracts depends on the affinity of the phytochemicals for the solvent. In the case under investigation, it was observed (Figure 2) that the extraction yield for pure acetone was higher than for pure ethanol and pure methanol. This suggests that the extraction yield increases with decreasing polarity of the solvent used in extraction. However, it was also found that the extraction yield for aqueous methanol was higher than for pure solvent, showing that in the case of methanol, increasing the water concentration is likely to promote the extraction of highly polar compounds. Total phenols exhibited higher solubility in acetone and in acetone aqueous solutions at high acetone content, while the further increase in water content contributed to reducing the extraction yields. In this experiment, at a fixed extraction time, temperature, and agitation speed, the total phenols yield of extraction from fresh TP ranged from 1.71 mg_GAE_/g_DM_ with pure methanol (run 10) to 3.13 ± 0.03 mg_GAE_/g_DM_ for acetone at 80% *v*/*v* (run 2). Therefore, further extraction experiments were carried out with acetone at 80% *v*/*v*. It must be remarked that the optimization of the extraction conditions from TP using organic solvents is outside the scope of the present work, and it has the sole purpose to provide an indication of the target values for the recovery of phenolic compounds in the liquors, as discussed in Section 3.3.1 (Biological Properties of Liquors).

### 3.3. Cellulose Isolation

The chemical hydrolysis process (Figure 1) applied to dried TP to obtain cellulose reached a yield of about 17.08 ± 0.44%. The HPH pretreatment contributed to obtaining a slightly but significantly (*p* < 0.05) higher cellulose yield of 18.60 ± 0.39%, showing how HPH pretreatment can be exploited to potentiate cellulose yield from tomato pomace by up to 9%. Moreover, HPH pretreatment also influenced cellulose morphology and structure, as well as the biological activity of the liquors obtained at the end of each chemical hydrolysis step.

#### 3.3.1. Biological Properties of Liquors

Based on Folin’s polyphenols determination (Figure 3), the acid hydrolysis step released higher amounts of total phenolic compounds than the successive alkaline hydrolysis and bleaching. Considering the total phenols quantification based on the 280-index method, the release yields obtained from acid hydrolysis were notably higher than the values estimated with the Folin method. However, it should be considered that the 280 nm reading is also used to estimate the sugar degradation products, which can be generated by the acid hydrolysis of the hemicellulose fraction [51,52], with a consequent overestimation of phenols content. Detoxification is, therefore, generally required to preserve the hemicellulose sugars removing the compounds that affect fermentation. The combination of the detoxification process with the recovery of phenols as high value-added compounds represents an interesting and widely investigated opportunity. Similar results were obtained in the evaluation of the reducing activity, with the acid liquor showing higher activity than the alkaline one, likely due to the preservation of phenolic compounds at acid conditions (Figure 3).

It must be highlighted that different treatment conditions were applied in preparing TP and HPH-TP samples, with the TP samples dried at 50 °C, while HPH-TP samples were suspended in water as received to be treated by HPH and then freeze-dried. However, the total phenolic content of the two samples, as measured in the extracts obtained with acetone (80% *v*/*v* in water), was not significantly different (*p* < 0.05), as shown in Figure 4. Moreover, also the reducing activity (FRAP values) did not show any significant (*p* < 0.05) difference (data reported in Appendix A). Therefore, it can be safely assumed that the different drying processes used to prepare TP or for HPH-TP treatment did not cause any significant degradation of the target bioactive compounds, as measured through Folin-Ciocalteau and FRAP assays.

Release yields of total phenols are significantly higher than the yields obtained from the same materials (TP and HPH-TP) through optimized solvent extraction with acetone at 80% *v*/*v* under agitation at 180 rpm at 25 °C for 24 h, as shown in Figure 4.

In particular, total phenols content extracted with acetone at 80% *v*/*v* was about 1.58 ± 0.02 mg_GAE_/g_DM_ and 1.82 ± 0.04 mg_GAE_/g_DM_ for TP and HPH-TP, respectively, whereas the significantly (*p* < 0.05) higher values of 10.55 ± 0.06 mg_GAE_/g_DM_ and 10.45 ± 0.10 mg_GAE_/g_DM_ for TP and HPH-TP, respectively, were obtained in the liquor from acid hydrolysis.

These findings suggest that the side streams of the lignocellulosic fractionation process (liquor from the different hydrolysis phases) can be exploited to efficiently recover phenolic compounds, replacing the conventionally applied solvent extraction step, hence contributing to enhancing the sustainability and economic viability of the cellulose recovery process from agri-food residues.

#### 3.3.2. Chemical and Morphological Characteristics of Isolated Cellulose

After the cascade chemical hydrolysis used to isolate cellulose from TP and HPH-TP (shown in Figure 1), the composition of the sample was determined through two-stage strong acid hydrolysis at high temperature and pressure, which was used to depolymerize cellulose, through cleavage of glycosyl units, to D-glucose, and hemicellulose to xylose, for spectrophotometric determinations (Table 3). The highest cellulose content and the lowest lignin and hemicellulose content were observed in HPH-TP, confirming that the HPH process contributed to improving the cellulose isolation process by de-structuring hemicellulose and lignin and by improving cellulose recovery from fragmented cell walls. This hypothesis is confirmed by previous studies, where different types of mechanical pretreatments are reported to contribute to enhancing cellulose yield by loosening fibril aggregation, breaking bonds between lamellae, and promoting defibrillation [53,54,55]. Consequently, it can be concluded that mechanical pretreatments, such as HPH, not only directly improve the deconstruction of hemicelluloses but also create favorable conditions for the subsequent chemical hydrolysis.

The acetic acid content was similar in both samples, as it is the main degradation product of xylan and, therefore, of hemicellulose. In this study, glucose and xylose have been evaluated to determine the content of cellulose and hemicellulose. Nevertheless, the obtained raw fibers also comprised galactose, galacturonic acid, aminogalactose, arabinose, and rhamnose, constituting the remaining sugar components, not reported in Table 3.

It must be remarked that, because of the intensive processing conditions in sample preparation, this analysis could lead to an overestimation of the cellulose and underestimation of hemicellulose content and, therefore, to an overestimation of the process efficiency in cellulose isolation.

FT-IR spectroscopy is an effective technique to investigate the chemical structure by identifying the functional groups of different materials and evaluating the structural changes that occurred during the applied treatments. For example, FT-IR was used to identify protein and starch changes caused by high hydrostatic pressure [56,57] and sonication or the effect of HPH in nanocellulose isolation [13]. Figure 5 clearly shows that FT-IR spectra of TP and HPH-TP exhibited common bands, which are typical of cellulose material, without any remarkable difference between the two samples. The spectrum of cellulose is characterized by absorption peaks between 950 and 1100 cm^−1^ because of the presence of C–O and C=C stretching vibration groups [58]. In particular, the peaks at 995 cm^−1^ [59,60], 1032 cm^−1^ [60,61], and 1055 cm^−1^ [62] are generally assigned to skeletal vibration involving C–O stretching, β-glycosidic linkages between sugar units. Based on Figure 4, it can be concluded that the cellulose isolated from HPH pretreated TP had a similar chemical structure as TP cellulose, suggesting that the chemical groups and conformation of the cellulose structure were not altered or destroyed by the mechanical treatment. Finally, it must be highlighted that no absorption peaks were detected at 1540 cm^−1^ (data not shown), suggesting that the samples did not contain any residual protein.

The morphology of the isolated cellulose after the chemical hydrolysis steps was observed using optical microscopy (Figure 6a,b) and SEM (Figure 6c–f). The cellulose fibers from TP appeared in the form of agglomerates of irregular morphology, where the original cell structure could be clearly detected. The cellulose fibers isolated from HPH-TP were, instead, completely different, with smaller agglomerates of irregular shape, including long needle-like debris with length from 600 to 950 µm and width from 10 to 30 µm. The significant variation in size and shape between TP and HPH-TP could be attributed to the fluid-mechanical stresses exerted by the HPH pretreatment, which improved the cellulose defibrillation and contributed to trimming down the length of the fibers. SEM analysis (Figure 6c–f) confirmed the optical microscopy observations. Cellulose from TP is characterized by a thickness of approximately 20 µm and is organized in individual sheets corresponding to the peel cell layers. In contrast, cellulose from HPH-TP exhibited a high defibrillation degree, with individual fibers separating from fiber bundles. Moreover, in HPH-TP cellulose, the sheets observed for TP are fragmented in small pieces (with an average size of about 100 µm) and a honeycomb-like structure characterized by large cavities and high void fraction. This structure ensures a significantly larger specific surface area than TP cellulose, improving the techno-functional properties of HPH-TP cellulose (which is sometimes referred to as fiber activation [63,64,65]). For example, HPH-activated TP fibers demonstrated a remarkable capability to act as stabilizers in Pickering emulsions [45].

Therefore, as can be clearly seen from Figure 7, the HPH pretreatment led to a strong defibrillation degree with long individual fibers.

Particle size distribution analysis showed that TP and HPH-TP isolated cellulose presented a unimodal distribution, with the main peaks located approximately at 630 μm and 315 μm, respectively (Figure 8). Moreover, the 90th percentile of TP and HPH-TP cellulose particles was around 1200 µm and 600 µm, respectively (Table 4). These results further suggest that HPH pretreatment was able to contribute to the disintegration of plant tissues, favoring the isolation of cellulose.

## 4. Conclusions

The present work shows for the first time that tomato pomace residues represent a viable and sustainable lignocellulosic source for cellulose isolation. Milder processing conditions than what is typically reported in the literature, based on chemical hydrolysis with sulphuric acid (4.7% *v*/*v*), followed by an alkali treatment with sodium hydroxide (4 N) contributed to efficiently remove the non-cellulosic constituents, resulting in a solid residue with a high content of D-glucose, hence suitable for cellulose isolation. Moreover, this work also demonstrated how the HPH mechanical pretreatment of tomato pomace was a viable option not only to increase the yields of cellulose but also to tailor the morphological properties of the isolated cellulose. The HPH pretreatment enabled obtaining a defibrillated cellulose with needle-shaped morphology and high surface area. These findings suggest that HPH pretreatment may support the development of more environmentally friendly cellulose isolation methods, enabling the reduction of the severity of the hydrolysis conditions. Moreover, the obtained results also suggest that through the implementation of a biorefinery approach, agri-food residues might be exploited to recover not only cellulose but also valuable bioactive compounds as side streams, with higher yields than with conventional solid-liquid extraction with organic solvents. Further studies need to be undertaken to further explore the proposed biorefinery approach through the optimized integration with the HPH pretreatment with milder chemical hydrolysis (e.g., optimizing treatment times and reactant concentration), as well as to produce high value-added nanocellulose.

## Figures and Tables

**Figure 1 foods-11-00266-f001:**
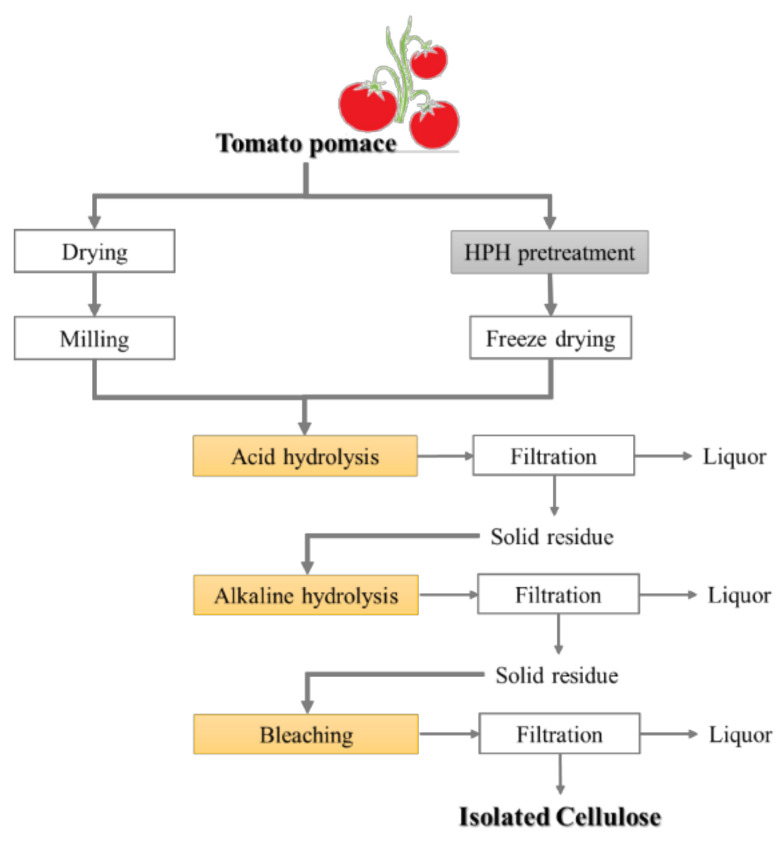
Schematic diagram of the procedures for extracting cellulose from tomato pomace (TP) and from tomato pomace pretreated by HPH (HPH-TP).

**Figure 2 foods-11-00266-f002:**
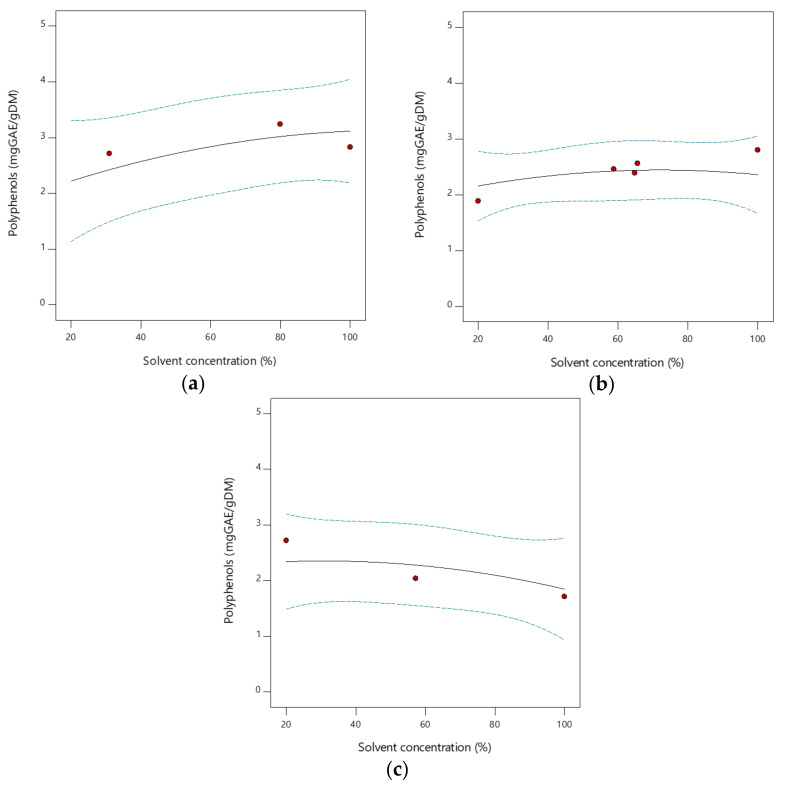
Dependence of response variable (total phenols, mg_GAE_/g_DM_) from solvent concentration (solid line) with the indication of the 95% confidence interval bands (dashed lines) and experimental design points (red circles), using (**a**) acetone, (**b**) ethanol and (**c**) methanol as organic solvents.

**Figure 3 foods-11-00266-f003:**
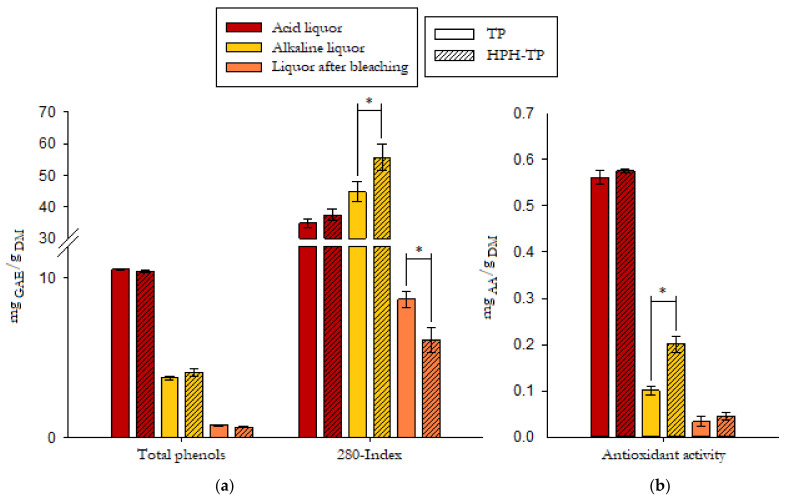
Total phenols (**a**) and reducing activity (**b**) in the liquor from acid and alkaline hydrolysis and bleaching of tomato pomace (TP) and high-pressure homogenized tomato pomace (HPH-TP). Values are reported as mean (*n* = 5) ± standard deviations. Asterisks denote statistically significant differences (*p* < 0.05).

**Figure 4 foods-11-00266-f004:**
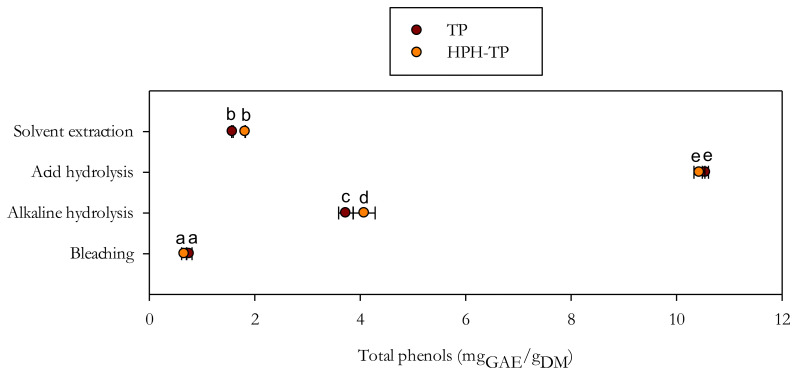
Comparison of total phenols content in liquors after the different stages of chemical hydrolysis, in comparison with conventional solid/liquid extraction with acetone (80% *v*/*v* in water). Different letters denote statistically significant (*p* < 0.05) differences.

**Figure 5 foods-11-00266-f005:**
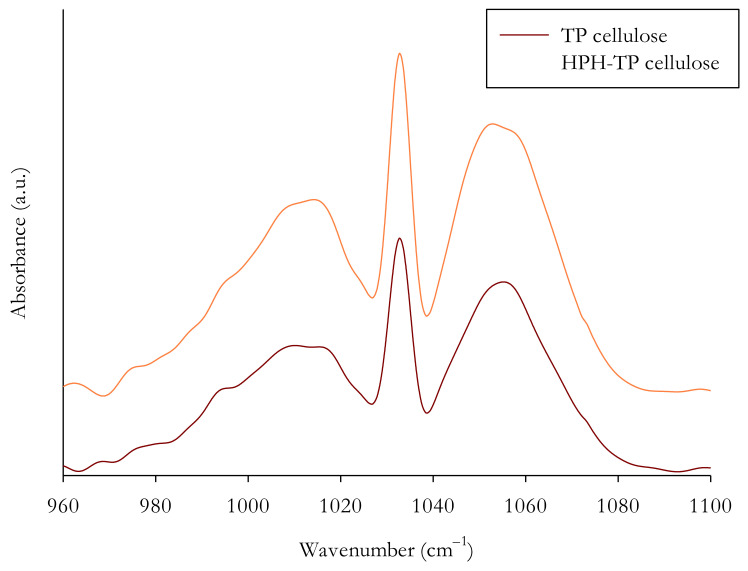
FT-IR absorbance spectra of the solid residues from TP (dark-red line) and HPH-TP samples (orange line), obtained after the cascade chemical treatments.

**Figure 6 foods-11-00266-f006:**
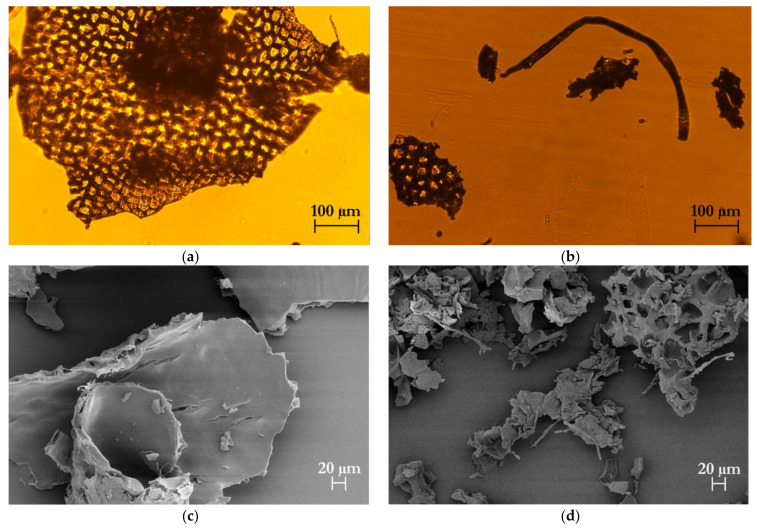
Optical microscopy (**a**,**b**) at 10× magnification and SEM (**c**–**f**) images at different magnifications (500× magnification for (**c**,**d**), 150× magnification for (**e**,**f**) of cellulose isolated from tomato pomace (TP) (left column, (**a**,**c**,**e**)) and high-pressure homogenized tomato pomace (HPH-TP) (right column, (**b**,**d**,**f**)) through the mild chemical hydrolysis cascade process.

**Figure 7 foods-11-00266-f007:**
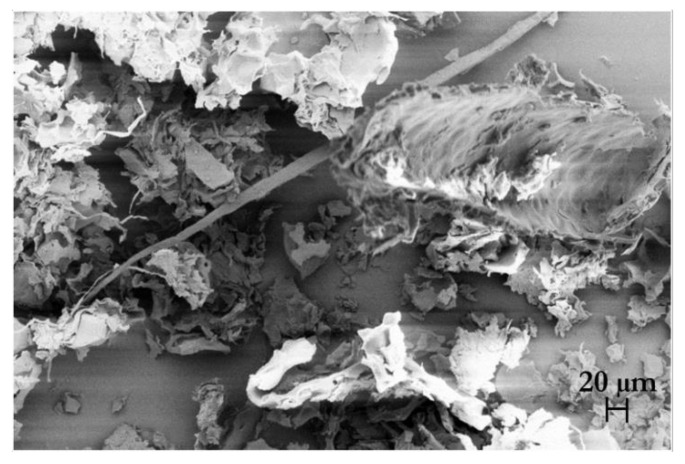
SEM micrographs at 300× magnification of cellulose extracted from HPH-TP.

**Figure 8 foods-11-00266-f008:**
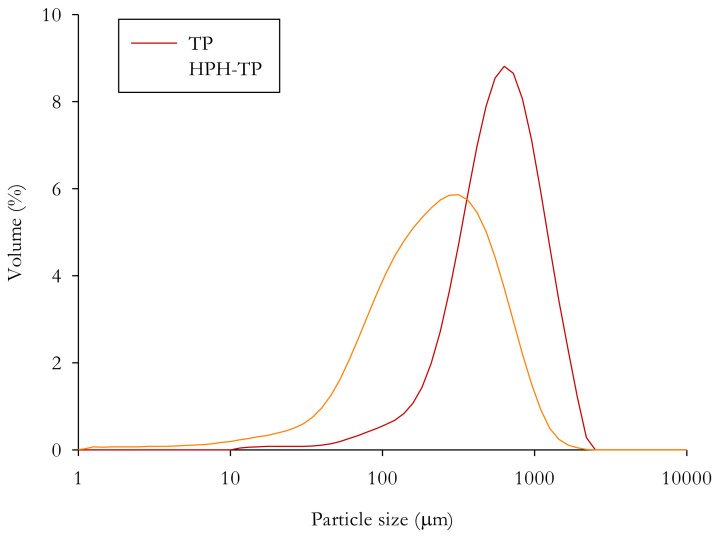
Particle size distribution of cellulose isolated from tomato pomace (TP) and high-pressure homogenized tomato pomace (HPH-TP) through the mild chemical hydrolysis cascade process.

**Table 1 foods-11-00266-t001:** The I-Optimal Design used for optimization of total phenols extraction.

Run Number	Type of Solvent	Solvent Concentration (% *v*/*v*)
1	Acetone	31
2	Acetone	80
3	Acetone	100
4	Ethanol	20
5	Ethanol	64.8
6	Ethanol	58.8
7	Ethanol	100
8	Ethanol	65.6
9	Methanol	20
10	Methanol	100
11	Methanol	57.2

**Table 2 foods-11-00266-t002:** Chemical composition of fresh tomato pomace (TP) and freeze-dried HPH-treated tomato pomace (HPH-TP).

Component	TP	HPH-TP
Moisture content (%)	80.70 ± 0.83 ^b^	6.94 ± 0.52 ^a^
Ash content (%_DM_)	3.47 ± 0.61 ^a^	3.26 ± 0.01 ^a^
Protein (%_DM_)	14.06 ± 1.04 ^b^	11.79 ± 0.61 ^a^
Fat (%_DM_)	1.20 ± 0.14 ^a^	1.43 ± 0.18 ^a^
Extractives (%_DM_)	8.24 ± 0.42 ^a^	14.66 ± 0.97 ^b^
Cellulose (%_DM_)	32.09 ± 2.24 ^a^	41.02 ± 1.79 ^b^
Hemicellulose (%_DM_)	26.62 ± 1.57 ^b^	6.27 ± 1.33 ^a^
Lignin (%_DM_)	30.54 ± 1.71 ^b^	24.90 ± 1.97 ^a^

Different letters in the same row denote statistically significant (*p* < 0.05) differences.

**Table 3 foods-11-00266-t003:** Lignocellulosic fractionation process of TP and HPH-TP in terms of cellulose, hemicellulose, and lignin content.

Component	TP	HPH-TP
Cellulose content (g/100 g_Fibre residue_)	29.33 ± 1.52 ^a^	37.72 ± 0.79 ^b^
Hemicellulose content (g/100 g_Fibre residue_)	18.74 ± 0.52 ^a^	4.62 ± 0.65 ^b^
Acetic acid content (g/100 g_Fibre residue_)	0.10 ± 0.02 ^a^	0.11 ± 0.02 ^a^
Acid soluble lignin (g/100 g_Fibre residue_)	0.22 ± 0.01 ^a^	0.18 ± 0.01 ^a^
Acid insoluble lignin (g/100 g_Fibre residue_)	0.52 ± 0.03 ^a^	0.48 ± 0.02 ^a^

Different letters in the same row denote statistically significant (*p* < 0.05) differences.

**Table 4 foods-11-00266-t004:** Characteristic percentile diameters (µm) of the cumulative size distribution of cellulose isolated from tomato pomace (TP) and high-pressure homogenized tomato pomace (HPH-TP) through the mild chemical hydrolysis cascade process.

Component	TP	HPH-TP
d(0,1)	21.6	55.7
d(0,5)	553.6	210.5
d(0,9)	1154.0	594.8

## Data Availability

Data is contained within the article (or Appendix A).

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
