# Peer review of "Cellulose Isolation from Tomato Pomace Pretreated by High-Pressure Homogenization"

_foods, 2022, doi:10.3390/foods11030266_

Round 1

Reviewer 1 Report

This paper addresses the use of high-pressure homogenization (HPH) to increase the extractability of cellulose from tomato waste. The subject matter addressed is timely and in keeping with the theme of a special issue of the journal.

The authors have written a very good introduction to the research they have undertaken, and have described the research methodology very well and in detail.

I provide comments below on the discussion of the results, which I believe should be expanded in some cases. However, the quality of the existing descriptions and presentation of results in most cases stands at a high level.

I have no comments on the Conclusions. In my opinion, they are well written, follow from the results of the experiments and contain research recommendations for the future.

The collection of literature cited are items from important journals, and were reasonably cited, mostly from the last few years.

Comments:

Table 1 – Why were only three tests performed for acetone and methanol, but six runs for ethanol (and twice at the same concentration 20%) ?

In section “Optimization of conventional extraction conditions” obtained interaction graphs should be better discussed. Each solvent gives different results when increasing its proportion in solution.

line 296- sentence has been cuted, please check

section 3.3. - The authors should discuss why HPH has increased cellulose yield

line 307-312- I recommend removing this text - it is a repetition of information given in the methodology.

Author Response

Responses to the comments of Reviewer 1

This paper addresses the use of high-pressure homogenization (HPH) to increase the extractability of cellulose from tomato waste. The subject matter addressed is timely and in keeping with the theme of a special issue of the journal.

The authors have written a very good introduction to the research they have undertaken, and have described the research methodology very well and in detail.

I provide comments below on the discussion of the results, which I believe should be expanded in some cases. However, the quality of the existing descriptions and presentation of results in most cases stands at a high level.

I have no comments on the Conclusions. In my opinion, they are well written, follow from the results of the experiments and contain research recommendations for the future.

The collection of literature cited are items from important journals, and were reasonably cited, mostly from the last few years.

R: We express our sincerest thanks to the reviewer for appreciating our work and for the constructive comments.

Comments:

Table 1 – Why were only three tests performed for acetone and methanol, but six runs for ethanol (and twice at the same concentration 20%)?

R: We thank the reviewer for pointing out this aspect. We mistakenly reported twice the run for EtOH at 20% w/w (only one run at this concentration was carried out). The 11 runs I-optimal were determined through randomized design by Design Expert for a full quadratic model in two factors, type of solvent (discrete variable) and solvent concentration (continuous variable). The I-optimal design usually places three runs at each vertex, and five runs in the center of the design region (which happens to be for ethanol). They were arranged in this order by the arbitrary choice of the authors, based on the consideration that ethanol has intermediate polarity (relative polarity of 0.654) with respect to acetone (0.355) and methanol (0.762) (according to Christian Reichardt, Solvents and Solvent Effects in Organic Chemistry, Wiley-VCH Publishers, 3rd ed., 2003). We have now modified the text accordingly.

In section “Optimization of conventional extraction conditions” obtained interaction graphs should be better discussed. Each solvent gives different results when increasing its proportion in solution.

R: We have now improved the discussion of interaction graphs in section 3.2 (Optimization of conventional extraction conditions) to better highlight the effects of solvent concentration on extraction yield. Now the manuscript reads (lines 296-308): “Under the same extraction conditions, e.g. time, temperature, agitation speed, sample particle size and solid-liquid ratio, the extraction efficiency is affected by the chemical nature of the phytochemicals, as well as the solvent used, for its polarity and concentration. However, in the case of vegetable biomass, the composition of the ex-tracts depends on the affinity of the phytochemicals for the solvent. In the case under investigation, it was observed (Figure 2) that the extraction yield for pure acetone was higher than for pure ethanol and pure methanol. This suggests that the extraction yield increases with decreasing polarity of the solvent used in extraction. However, it was also found that the extraction yield for aqueous methanol was higher than for pure solvent, showing that in the case of methanol, increasing the water concentration likely enhances the extraction yield of highly polar compounds. Total phenols exhibited higher solubility in acetone and in acetone aqueous solutions at high acetone content, while the increase in water content contributed to reducing the extraction yields.”

line 296- sentence has been cuted, please check

R: We have now corrected the text.

section 3.3. - The authors should discuss why HPH has increased cellulose yield

R: We thank the reviewer for the suggestion. We have now improved the discussion and we have added new literature references to highlight how mechanical pretreatments may increase cellulose yield. Now the manuscript reads at lines 375-384: “The highest cellulose content and the lowest lignin and hemicellulose content was observed in HPH-TP, confirming that the HPH process contributed to improving the cellulose isolation process by de-structuring hemicellulose and lignin and by improving cellulose recovery from fragmented cell walls. This hypothesis is confirmed by previous studies, where different types of mechanical pretreatments are reported to contribute to enhancing cellulose yield by loosing fibril aggregation, breaking bonds between lamellae and promoting defibrillation [52–54]. Consequently, it can be concluded that mechanical pretreatments, such as HPH, not only directly improve the deconstruction of hemicelluloses, but also create favorable conditions for the subsequent chemical hydrolysis.”

line 307-312- I recommend removing this text - it is a repetition of information given in the methodology.

R: We thank the reviewer for the suggestion. We have removed the text.

Reviewer 2 Report

The article is very interesting and well written, but it contains some inaccuracies that affect its quality.

I do not understand why the authors compare TPC and FRAP in solutions obtained from two differently prepared materials (drying 50 degrees vs freeze-drying) and are discussing the effect of the extractant used based on that. It is obvious that using the dried material as raw material comparing to freeze-dried - the  TPC and FRAP values will differ from the beginning.
This is subject to a methodological error, because it is known that the temperature affects the oxidation of polyphenols, and additionally, the results of FRAP (besides, please do not use the term antioxidant activity when using the FRAP method, rather reducing activity).
Therefore, what additional information is to be provided by the measurement of TPC and FRAP, if the purpose of the article is to obtain cellulose.

I recommend removing it from the manuscript.

Line 119, please use SI metric system, Torr is not a SI unit

Author Response

Responses to the comments of Reviewer 2

The article is very interesting and well written, but it contains some inaccuracies that affect its quality.

R: We thank the reviewer for the positive comments and for the frank constructive criticism.

I do not understand why the authors compare TPC and FRAP in solutions obtained from two differently prepared materials (drying 50 degrees vs freeze-drying) and are discussing the effect of the extractant used based on that. It is obvious that using the dried material as raw material comparing to freeze-dried - the TPC and FRAP values will differ from the beginning.

This is subject to a methodological error, because it is known that the temperature affects the oxidation of polyphenols, and additionally, the results of FRAP (besides, please do not use the term antioxidant activity when using the FRAP method, rather reducing activity).

Therefore, what additional information is to be provided by the measurement of TPC and FRAP, if the purpose of the article is to obtain cellulose.

I recommend removing it from the manuscript.

R: We agree with the reviewer that we compared TPC and FRAP in solutions obtained from two differently prepared materials. Nevertheless, as shown by the data reported in Figure 4, the TPC of the two materials subjected to different drying conditions (TP and HPH-TP) is not significantly different when subjected to solvent extraction (extraction in acetone 80% v/v). We have now carried out an additional characterization of the FRAP value of the extracts under the same conditions from TP and HPH-TP, showing that also for this parameter there are no significant differences (the additional results have now been reported in the Supplementary Material, Table S2). Therefore, we can safely assume that the different drying processes used to prepare TP or for HPH-TP treatment did not cause any significant degradation of the bioactive compounds (drying conditions of TP were set at 50°C to avoid degradation of thermolabile compounds, freeze-drying was used for HPH-TP as this sample is in an aqueous suspension). We have now better specified this in the manuscript at lines 339-348.

About the second part of the comment, we consider the information about the biological activity of the liquors (as characterized by TPC and FRAP) to be important for the valorization of the side streams in the recovery of cellulose, contributing to improving the process sustainability. This has been better explained in the manuscript, which now reads (lines 365-369) “These findings suggest that the side streams of the lignocellulosic fractionation process (liquor from the different hydrolysis phases) can be exploited to efficiently recover phenolic compounds, replacing the conventionally applied solvent extraction step, hence contributing to the enhancing the sustainability and economic viability of the cellulose recovery process from agri-food residues.”

Line 119, please use SI metric system, Torr is not a SI unit

R: The change from Torr to Pa has been made accordingly.

Reviewer 3 Report

The manuscript presents valuable information on the valorisation of tomato pomace to recover cellulose and is of great interest considering the circular economy concept. However, the manuscript structure is difficult to understand, the optimization is not clearly described and justified and the results are not discussed very well. Please see the detailed remarks below: 

Abstract: The aim and the methods used should be mentioned.

The introduction is well written and supports the study.

Methods: Section 2.5: more details about the design used are recommended to be included (e.g. model evaluation, number of replicates, blocks, central points (if applicable), etc.). Furthermore, the authors mention that there are 2 factors, but in the results, there are presented graphics with only one factor (concentration). Please clarify if you used one optimization for each solvent (this I understand from the results). In addition, no information about the optimization is provided: have you used desirability approach? Which were the constraints?

L193: Please avoid starting a sentence with a number. Check in the whole manuscript.

Results: Section 3.2: the optimization results are incomplete. Please provide the mathematical models used (maybe in the methods section) and provide the table including the coefficients and the significance of the terms (see this article as guidance https://doi.org/ 10.3390/app11125403). It is not clear if two factors were used: if so, a 3D plot should be provided. Please clarify how have you obtained an optimal value for each solvent (have you put the constraint to keep a certain solvent, or have you performed a different optimization for each solvent?). Why have the authors optimized the extraction of polyphenols and not the extraction of cellulose which is the subject of the manuscript?  

Section 3.3: Where are these data presented (table, figure)?

Figure 6: The magnification for each image should be provided. Also g and h images: why do you compare 200 to 20 um? the same magnification should be used for both samples. 

Figure 7: Please remove the table from the figure and put the data separately. 

Generally, the discussion of the results is poor. More discussions and comparisons with other results from the literature are needed.

Author Response

Responses to the comments of Reviewer 3

The manuscript presents valuable information on the valorisation of tomato pomace to recover cellulose and is of great interest considering the circular economy concept. However, the manuscript structure is difficult to understand, the optimization is not clearly described and justified and the results are not discussed very well. Please see the detailed remarks below:

Abstract: The aim and the methods used should be mentioned.

R: The abstract has been modified, following the suggestion of the reviewer.

The introduction is well written and supports the study.

R: We thank the reviewer for appreciating our work and for the positive comments.

Methods: Section 2.5: more details about the design used are recommended to be included (e.g. model evaluation, number of replicates, blocks, central points (if applicable), etc.). Furthermore, the authors mention that there are 2 factors, but in the results, there are presented graphics with only one factor (concentration). Please clarify if you used one optimization for each solvent (this I understand from the results). In addition, no information about the optimization is provided: have you used desirability approach? Which were the constraints?

R: We agree with the reviewer, and, therefore, we have now provided the required details in the Supplementary Materials, which have now been added to the submission. The optimization was performed by using two experimental factors. This is described more in detail in the following points.

L193: Please avoid starting a sentence with a number. Check in the whole manuscript.

R: The suggested changes have been made accordingly.

Results: Section 3.2: the optimization results are incomplete. Please provide the mathematical models used (maybe in the methods section) and provide the table including the coefficients and the significance of the terms (see this article as guidance https://doi.org/10.3390/app11125403). It is not clear if two factors were used: if so, a 3D plot should be provided. Please clarify how have you obtained an optimal value for each solvent (have you put the constraint to keep a certain solvent, or have you performed a different optimization for each solvent?). Why have the authors optimized the extraction of polyphenols and not the extraction of cellulose which is the subject of the manuscript?

R: We agree with the reviewer that the aim of the manuscript is the extraction of cellulose, rather than the optimization of phenols extraction. The optimization of the extraction of total phenols was motivated simply by the need to compare the non-optimized liquor composition with the optimized solvent extraction, to show how advantageous the valorization of these side-streams can be. We have tried to better clarify this aspect in the manuscript now in lines 312-315, which now reads: “It must be remarked that the optimization of the extraction conditions from TP using organic solvents is outside the scope of the present work, and it has the sole purpose to provide an indication of the target values for the recovery of phenolic compounds in the liquors, as discussed in section 3.3.1 (Biological properties of liquors).”

The optimization was performed by using two experimental factors, namely the type of solvent, which is a discrete variable, and solvent concentration, which is a continuous variable. By setting the type of variable (discrete and continuous) in the software automatically set a constrain to very the concentrations for each solvent. Therefore, we have reported the results of the RSM in three different graphs for each discrete variable used. However, in response to the comment of the reviewer, we have now reported the 3D contour plots in the Supplementary Material. This type of plot should be interpreted considering that the surface in-between two solvents corresponds to the behavior of a hypothetical solvent with intermediate properties.

We have also added a more detailed description of the design and optimization procedure and results in the Supplementary Materials, including the mathematical models used and a table (Table S1) with the coefficients and the significance of the terms.

Section 3.3: Where are these data presented (table, figure)?

R: Considering that the data to be presented are only two, namely the cellulose yield for TP and for HPH-TP, we assessed as not necessary to add a table or figure. However, after the comment of the reviewer, we have now reported these data together with their standard deviation.

Figure 6: The magnification for each image should be provided. Also g and h images: why do you compare 200 to 20 um? The same magnification should be used for both samples.

R: We agree with the reviewer and we decided to remove the panels g and h from Figure 6. Image in Figure 6h has been added as Figure 7, as we considered to provide valuable morphological information about the effect of HPH at higher magnification. The information about magnification has been added in the figures' legends.

Figure 7: Please remove the table from the figure and put the data separately.

R: As suggested by the reviewer, we have now removed the table from the figure.

Generally, the discussion of the results is poor. More discussions and comparisons with other results from the literature are needed.

R: We thank you for the insightful comment. We have now improved the discussion, adding also the comparison with literature data (please see lines 296-308, 312-315, 339-348, 367-369, and 378-384).

Round 2

Reviewer 2 Report

The Authors have introduced thoroughly all the recommended changes. The manuscript is now coherent and self-explanatory, as well well discussed. 

Reviewer 3 Report

The manuscript was improved according to the suggestions.